# The SII-PNI score: A novel composite biomarker for personalized mortality risk prediction in peritoneal dialysis patients

**Min Zhou**◉©, **Guoyi Wang**©, **Ping Yue**, **Hua Lin**, **Min Chen**, **Xuan Chen**, **Jinwen Zhao***, **Yong Xu***

Department of Nephropathy, The Affiliated Huai'an NO.1 People's Hospital of Nanjing Medical University, Huai'an, China

© These authors contributed equally to this work.
* zjwhayy@163.com (JZ); haeyxy@163.com (YX)

## Abstract

### Objective

Accurate risk stratification is crucial for personalized management in peritoneal dialysis (PD). This study aimed to develop and validate a novel biomarker, the SII-PNI score, integrating systemic immunoinflammatory index (SII) and prognostic nutritional index (PNI), for personalized mortality risk prediction in PD patients.

### Methods

A retrospective cohort study analyzed data from 330 patients initiating PD between December 2005 and June 2023 at a single center. Patients were stratified into four risk groups based on median cut-off values for SII and PNI. The association between SII-PNI risk groups and mortality (all-cause, cardiovascular [CVD], infection-related) was assessed using Kaplan-Meier survival analysis and multivariable Cox proportional hazard models. The predictive performance of the SII-PNI score was evaluated using receiver-operating characteristic (ROC) curves and compared to individual components (SII, PNI) and CRP. Random survival forests assessed variable importance.

### Results

The high-risk group (G4: high SII + low PNI) had the shortest PD duration, highest mortality, and worst survival outcomes. Compared to the low-risk group (G3: low SII + high PNI), G4 had significantly increased risks of all-cause mortality (adjusted HR 3.36, 95% CI 1.93–8.67), CVD mortality (adjusted HR 3.74, 95% CI 2.41–19.60), and infection mortality (adjusted HR 4.32, 95% CI 2.58–20.4) in fully adjusted models. The SII-PNI score demonstrated superior predictive ability (AUC: all-cause 0.80,

**Data availability statement:** All relevant data are within the manuscript and its Supporting Information files.

**Funding:** This research was supported by Huai'an Municipal Health Commission Research Project (Grant number HAWJ2024006), Huai'an, China. There was no additional external funding received for this study. The funders had no role in study design, data collection and analysis, decision to publish, or preparation of the manuscript.

**Competing interests:** The authors have declared that no competing interests exist.

CVD 0.80, infection 0.81) compared to SII, PNI, or CRP alone. Random survival forests confirmed the critical importance of the individual components (platelets, neutrophils, lymphocytes, albumin) for outcomes.

## Conclusions

The SII-PNI score, derived from readily available blood parameters, is a powerful and convenient tool for personalized risk stratification in PD. Patients identified as high-risk warrant intensified monitoring and early interventions. This composite biomarker represents a significant step towards personalized and precision medicine in PD care, with potential for implementation in routine clinical practice using standard laboratory data.

## 1. Introduction

Peritoneal dialysis (PD) is a well-established kidney replacement therapy in patients with end-stage kidney disease (ESKD). Despite technological and management advances, the mortality remains high, driven significantly by inflammation and malnutrition [1]. Chronic microinflammation, fueled by pro-inflammatory cytokines and oxidative stress [2], and malnutrition, arising from reduced intake, peritoneal albumin losses and inflammation itself [3], are strongly linked to adverse cardiovascular and infection outcomes, culminating in poor prognosis [4]. Consequently, inflammation and nutritional status are critical prognostic factors in PD [5–8].

The pursuit of personalized medicine demands accurate, practical tools for risk stratification to guide tailored management. While individual biomarkers like C-reactive protein (CRP) have been studied, they often lack robustness or comprehensiveness. Composite indices integrating multiple parameters offer greater clinical value by providing a more holistic view of the patient's immune-inflammatory-nutritional axis. The systemic immunoinflammatory index (SII), calculated from platelet, neutrophil and lymphocyte counts, is a novel marker reflecting systemic inflammation and immune balance. It demonstrates superior predictive value compared to CRP, neutrophil-to-lymphocyte ratio (NLR), monocyte-to-lymphocyte ratio (MLR), and platelet-to-lymphocyte ratio (PLR) [9,10]. The prognostic nutritional index (PNI), combining serum albumin and total lymphocyte count [11], is a well-established objective indicator of protein-energy wasting and malnutrition [12], providing valuable prognostic information in PD patients, often outperforming other nutritional indices like the Geriatric Nutritional Risk Index (GNRI) in simplicity and utility [13].

The combination of SII and PNI offers a unique opportunity to simultaneously capture two inextricably linked pathophysiological axes: systemic inflammation and nutritional status. In ESKD, inflammation and malnutrition are intertwined in a vicious cycle that synergistically drives adverse outcomes, and the SII-PNI score holistically captures this interplay. While their combined predictive value has been explored in oncology [14–16], its application and significance in PD, particularly for personalized

risk prediction, remain unexplored. Therefore, this study aimed to develop and validate the SII-PNI score (operationalized as risk groups based on SII and PNI cut-offs) as a novel, easily calculable composite biomarker and investigate its utility for personalized mortality risk stratification in patients undergoing PD.

## 2. Methods

This was a retrospective, observational study and carried out at a single PD center of Nanjing Medical University Affiliated Huai'an NO.1 People's Hospital. This study used pre-existing clinical data from Huai'an NO.1 People's Hospital. Although initial data processing required temporary access to identifiable information (medical record numbers, dates of hospitalization and names) for data linkage, all direct identifiers were permanently removed before analysis and replaced with anonymous codes. The resulting dataset was fully anonymized with no means of tracing individuals. The de-identified data were stored on a password-encrypted server and accessible solely to the principal investigator. This protocol was approved by the Medical Research Ethics Committee of Huai'an NO.1 People's Hospital (Approval number KY-2024-320-01), the requirement for informed consent was formally waived based on the retrospective design and use of irreversibly de-identified data. All methods were performed in accordance with the relevant guidelines and regulations (e.g., Declaration of Helsinki). Data confidentiality was strictly maintained.

### 2.1. Study population

The study cohort comprised incident patients who received PD catheterization and initiated continuous ambulatory PD (CAPD) as their first kidney replacement therapy for ESKD between December 1, 2005, and June 30, 2023. Inclusion criteria were: age ≥ 16 years and survival on PD for at least 3 months. Exclusion criteria were: history of hemodialysis (HD) or kidney transplantation; PD initiation for acute kidney injury or acute heart failure; presence of active infectious diseases, blood system diseases, immunosuppressive drug use before PD start, or malignant tumors; and missing essential outcomes or clinical data. Details of the recruitment process were shown in Fig 1.

### 2.2. Data collection and measurements

To enhance representativeness and minimize missing data, baseline was defined as the average of values recorded within 6 months after PD initiation. Blood samples were collected on the second day of hospitalization and analyzed in a standardized laboratory. Clinical data extracted from electronic medical records included: demographics (age, gender), anthropometrics (body mass index, BMI), comorbidities (diabetes mellitus-DM, hypertension-HP, cardiovascular disease-CVD), PD related factors (duration, catheterization method), laboratory parameters (globulin, platelets, neutrophils, leukocytes, monocytes, lymphocytes, creatinine, urea nitrogen, serum albumin, cholesterol, HDL-C, LDL-C, serum sodium, calcium, potassium, parathyroid hormone, etc.). Data for research purposes were accessed and extracted between November 1, 2024, and January 31, 2025. All patients received standardized PD training from qualified nurses and passed competency assessments. All received CAPD using Baxter Healthcare (Guangzhou, China) solutions (Dianeal 1.5% or 2.5% dextrose) and twin-bag Y-set systems.

### 2.3. Follow-up and endpoints

Patients were followed up routinely every six months. The observation period spanned from PD initiation until death, transfer to HD, kidney transplantation, or the end of follow-up (June 30, 2023). The primary endpoint was all-cause mortality, secondary endpoints were CVD and infection-associated mortality. CVD death was attributed to coronary events, cardiomyopathy, cardiac dysrhythmia, sudden cardiac death, congestive heart failure, ischemic brain injury, cerebrovascular accident, or peripheral vascular disease. For out-of-hospital deaths, causes were determined via telephone interviews with family members.

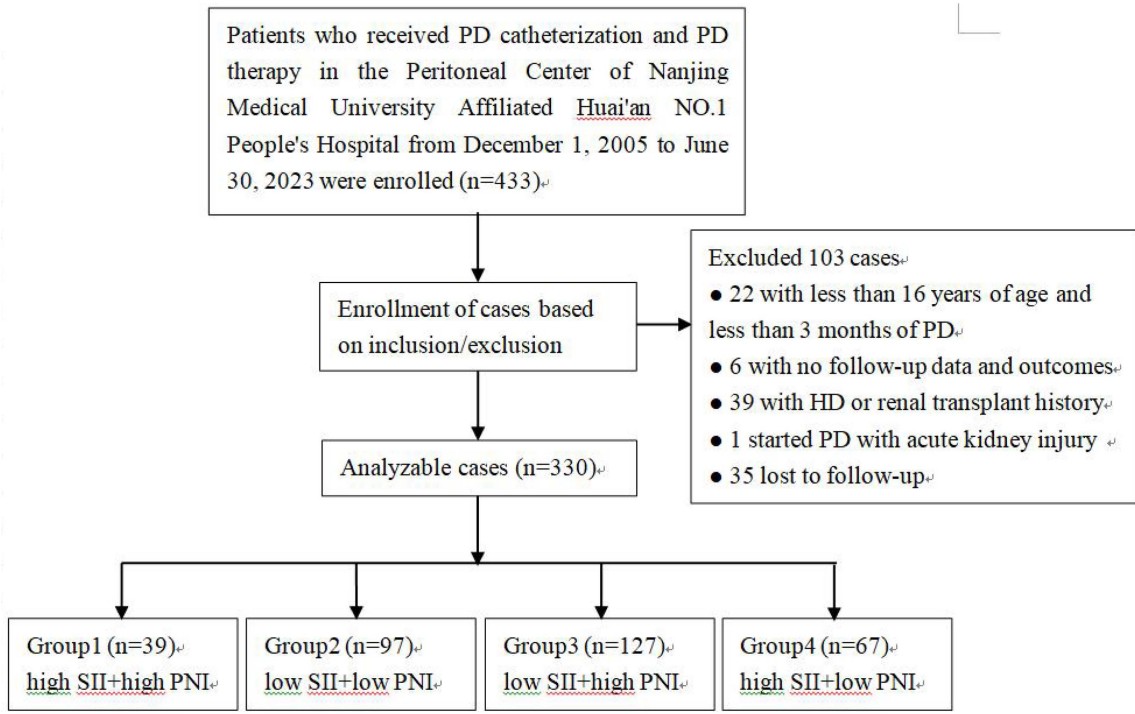

**Fig 1. Flowchart of participants inclusion.** PD: peritoneal dialysis; HD: hemodialysis.

## 2.4. Calculation of the indices and patient grouping

SII = platelet count ($10^9$/L) × neutrophil count ($10^9$/L) / lymphocyte count ($10^9$/L); PNI = serum albumin (g/L) + 5 × lymphocyte count (×$10^9$/L).

Cut-off values were determined based on cohort medians: SII high (≥791), low (<791); PNI high (≥46), low (<46). Patients were stratified into four risk groups:

Group 1 (G1): high SII + high PNI (Intermediate Risk)

Group 2 (G2): low SII + low PNI (Intermediate Risk)

Group 3 (G3): low SII + high PNI (Low Risk)

Group 4 (G4): high SII + low PNI (High Risk)

We chose the median as it is an objective, data-drivencut-offs that equally split the cohort, allowing for initial risk stratification. This approach is commonly used in exploratory biomarker studies, especially in the absence of universally established clinical thresholds for SII and PNI in PD populations.

## 2.5. Statistical analysis

Data were presented as numbers (percentages), means±standard deviation (SD), or medians (interquartile range, IQR) based on distribution. SPSS 27.0 (IBM, USA) was used. Group comparisons employed Student's t-test, Mann-Whitney U test, or Chi-square test (with continuity correction or Fisher's exact test) as appropriate. Survival differences were analyzed using Kaplan-Meier curves (Log-rank test). Cox proportional hazards regression models assessed hazard ratios

(HR) and 95% confidence intervals (CI) for associations between SII-PNI risk groups and mortality. Four models were constructed, Crude: No adjustments; Model 1: adjusted for age, gender, DM, HP, CVD, BMI and catheterization method; Model 2: Model 1 + lymphocyte count; Model 3: Model 2 + triglyceride. We adjusted for lymphocyte count to determine if the SII-PNI score provides prognostic value beyond its cellular components, further adjusted for triglycerides, a metabolic parameter closely linked to both inflammation and nutrition in ESKD, to test the independence of the SII-PNI score from related metabolic disturbances. The proportional hazards assumption for all Cox models were verified using Schoenfeld residuals, with no significant violations detected (all $P$-values> 0.05). Receiver-operating characteristic (ROC) curves evaluated the diagnostic ability of SII-PNI and other markers (SII, PNI, CRP) for mortality prediction. Random survival forests, an ensemble tree method for right-censored survival data, determined the relative importance of clinical predictors. $P$-values< 0.05 were considered significant.

## 3. Results

### 3.1. Baseline characteristics of the participants

A total of 330 patients were enrolled in this study, baseline characteristics stratified by SII-PNI risk groups were detailed in Table 1. Median SII and PNI values were 791 and 46 respectively, patient distribution was: G1 (n = 39), G2 (n = 97), G3 (n = 127), G4 (n = 67). The median age was 41.03 ± 12.94 years, with 42.73% (n = 141) females. The high-risk group (G4) exhibited distinct characteristics: higher BMI, shorter disease course, predominance of laparotomy catheter insertion (73.13%), highest prevalence of baseline CVD (44.78%), shortest PD duration, lowest lymphocyte counts and serum albumin levels, and the highest number of all-cause deaths (n = 20, 29.85%). Cardiac structure and function were also worse in G4, with increased left ventricular end-diastolic diameter (LVEDD), interventricular septum thickness (IVST), left ventricular posterior wall thickness (LVPWT) and reduced left ventricular ejection fraction (LVEF) compared to other groups, particularly the low-risk group (G3).

### 3.2. Survival analysis and Cox regression analysis

Over a median follow-up of 50.51 ± 38.69 months, 41 patients (12.42%) died, including 13 CVD deaths and 9 infection-related deaths. Refractory fungal peritonitis, pneumonia, and sepsis were the leading infection causes; heart failure, coronary disease, cerebral hemorrhage, and malignant arrhythmia caused CVD deaths.

Risk stratification based on SII-PNI was highly predictive of outcomes. Cumulative incidence curves (Fig 2) showed G4 had the highest incidence of all-cause, CVD, and infection mortality. KM survival curves (Fig 3) demonstrated stark contrasts, G3 (low risk) had the best survival probability, while G4 (high risk) had the worst survival probability for all endpoints, G1 and G2 (intermediate risk) showed intermediate survival. Cox regression analysis further confirmed the strong, independent association between the high-risk SII-PNI group (G4) and mortality, even after extensive adjustment for confounders (Table 2). Compared to the low-risk group (G3), G4 had significantly increased risk in all models. Especially in the fully adjusted Model 3, G4 had elevated all-cause death (HR 3.36, 95% CI 1.93–8.67, $p < 0.001$), CVD death (HR 3.74, 95% CI 2.41–19.60, $p < 0.001$), and infection-related death (HR 4.32, 95% CI 2.58–20.4, $p < 0.001$).

### 3.3 Diagnosis ability of SII-PNI score in PD patients

ROC analysis demonstrated that SII-PNI score showed superior discriminative power for all-cause mortality compared to SII or PNI alone (AUC: 0.80 vs. 0.58 and 0.58, respectively), confirming the composite index provides synergistic prognostic value (Fig 4a). The same applied to CVD and infection-associated mortality (Fig 4b-c). Diagnostic test statistics for SII-PNI predicting all-cause mortality showed an optimal cut-off score of 0.85 (sensitivity 0.70, specificity 0.85), with a high negative predictive value (0.93) and a C-index of 0.92, indicating excellent discriminative ability (Table 3).

**Table 1. Baseline characteristics of the study participants.**

| Clinical Characteristics | Total (n = 330) | G1 (n = 39) | G2 (n = 97) | G3 (n = 127) | G4 (n = 67) | *P*-value |
|---|---|---|---|---|---|---|
| **Baseline indicators** | | | | | | |
| Age (years) | 41.03±12.94 | 37.40±11.00 | 42.70±12.40 | 41.70±12.50 | 39.40±15.00 | 0.045 |
| Sex | | | | | | 0.727 |
| Male (n, %) | 189 (57.27) | 22 (56.41) | 60 (61.86) | 68 (53.54) | 39 (58.21) | |
| Female (n, %) | 141 (42.73) | 17 (43.59) | 37 (38.14) | 59 (46.46) | 28 (41.79) | |
| BMI (kg/m$^2$) | 22.71±3.63 | 21.80±2.82 | 22.60±2.79 | 22.50±3.74 | 23.70±4.65 | 0.177 |
| Course of disease (months) | 21.62±36.54 | 19.30±37.30 | 24.20±44.50 | 23.40±33.80 | 15.90±27.20 | 0.180 |
| Dialysis duration (months) | 50.51±38.69 | 46.30±34.50 | 45.50±33.30 | 57.60±33.60 | 39.10±36.60 | 0.005 |
| Catheterization(n, %) | | | | | | 0.047 |
| Laparotomy | 211 (63.94) | 28 (71.79) | 64 (65.98) | 70 (55.12) | 49 (73.13) | |
| Seldinger | 119 (36.06) | 11 (28.21) | 33 (34.02) | 57 (44.88) | 18 (26.87) | |
| All-cause Deaths (n, %) | 41 (12.42) | 6 (15.38) | 9 (9.28) | 6 (4.72) | 20 (29.85) | <0.001 |
| **Kidney failure causes** | | | | | | 0.953 |
| CGN (n, %) | 293 (88.79) | 37 (94.87) | 88 (90.72) | 109 (85.83) | 59 (88.06) | |
| DN (n, %) | 21 (6.36) | 1 (2.56) | 7 (7.22) | 9 (7.09) | 4 (5.97) | |
| HN (n, %) | 4 (1.21) | 0 (0) | 1 (1.03) | 2 (1.57) | 1 (1.49) | |
| Others (n, %) | 12 (3.64) | 1 (2.56) | 1 (1.03) | 7(5.51) | 3(4.48) | |
| **Complications** | | | | | | |
| DM (n, %) | 34 (10.30) | 1 (2.56) | 12 (12.37) | 13 (10.24) | 8 (11.94) | 0.354 |
| HP (n, %) | 252 (76.36) | 27 (69.23) | 79 (81.44) | 100 (78.74) | 46 (68.66) | 0.165 |
| CVD (n, %) | 69 (20.91) | 6 (15.38) | 15 (15.46) | 18 (14.17) | 30 (44.78) | <0.001 |
| **Laboratory indicators** | | | | | | |
| White blood count (*10$^9$/L) | 6.40±1.76 | 5.93±1.63 | 6.52±1.97 | 6.57±1.73 | 6.21±1.51 | 0.129 |
| Hemoglobin (g/l) | 100.90 ±18.72 | 102.00±20.20 | 98.60±18.50 | 101.00±18.00 | 103.00±19.40 | 0.497 |
| Platelet count (*10$^9$/L) | 189.36±75.26 | 175.00±69.300 | 187.00±76.600 | 194.0±70.90 | 169.0±79.60 | 0.289 |
| Neutrophil count (*10$^9$/L) | 4.58±1.79 | 4.31±1.41 | 4.70±2.12 | 4.85±1.72 | 4.11±1.58 | 0.093 |
| Lymphocyte count (*10$^9$/L) | 1.21±0.91 | 1.11±0.45 | 1.09±0.39 | 1.13±0.47 | 1.02±0.43 | <0.001 |
| Monocyte count (*10$^9$/L) | 0.43±0.22 | 0.41±0.22 | 0.42±0.18 | 0.46±0.25 | 0.41±0.19 | 0.439 |
| Fibrinogen (g/l) | 4.43±1.39 | 4.15±1.34 | 4.28±1.51 | 4.60±1.37 | 4.48±1.29 | 0.17 |
| CRP (mg/l) | 22.21±18.99 | 24.10±18.60 | 22.70±20.40 | 19.80±17.70 | 25.40±19.9 | 0.528 |
| Calcium (mmol/l) | 2.29±0.22 | 2.33±0.19 | 2.26±0.23 | 2.29±0.23 | 2.30±0.22 | 0.323 |
| Phosphate (mmol/l) | 1.65±0.46 | 1.63±0.46 | 1.67±0.48 | 1.68±0.49 | 1.56±0.39 | 0.419 |
| PTH (Pg/ml) | 306.11±252.26 | 310±292 | 302.0±251.0 | 305.0±253.0 | 312.0±234.0 | 0.895 |
| ALP (u/l) | 101.58±87.15 | 114.0±98.60 | 94.80±56.30 | 106.0±115.0 | 95.10±49.60 | 0.869 |
| Total protein (g/l) | 64.91±6.65 | 61.50±7.17 | 63.0±5.93 | 65.80±6.22 | 59.20±6.87 | <0.001 |
| Albumin (g/l) | 39.95±4.63 | 40.70±4.75 | 39.30±4.61 | 42.10±3.15 | 37.40±4.68 | <0.001 |
| Total bilirubin (umol/l) | 6.30±2.46 | 7.04±3.82 | 6.07±2.34 | 6.32±2.06 | 6.17±2.29 | 0.629 |
| ALT (u/l) | 18.16±10.32 | 20.40±14.40 | 17.60±9.44 | 18.0±9.57 | 17.90±10.20 | 0.963 |
| AST (u/l) | 18.28±7.47 | 17.90±8.13 | 18.10±7.54 | 18.80±7.87 | 17.70±6.19 | 0.819 |
| HDL-C (mmol/l) | 1.37±0.46 | 1.36±0.61 | 1.44±0.50 | 1.33±0.42 | 1.32±0.38 | 0.355 |
| LDL-C (mmol/l) | 2.55±0.78 | 2.40±0.74 | 2.52±0.78 | 2.57±0.73 | 2.63±0.90 | 0.525 |
| Cholesterol (mmol/l) | 4.45±1.10 | 4.44±1.15 | 4.44±1.19 | 4.38±0.95 | 4.59±1.21 | 0.959 |
| Triglyceride (mmol/l) | 1.69±0.86 | 1.94±1.01 | 1.43±0.56 | 1.71±0.90 | 1.88±0.97 | 0.005 |
| Blood glucose (mmol/l) | 5.40±1.74 | 5.55±1.89 | 5.36±1.85 | 5.31±1.57 | 5.52±1.85 | 0.668 |
| BUN (mmol/l) | 21.97±7.32 | 22.2±6.99 | 21.90±6.96 | 23.0±8.58 | 19.80±4.64 | 0.143 |

*(Continued)*

**Table 1.** (Continued)

| Clinical Characteristics | Total (n = 330) | G1 (n = 39) | G2 (n = 97) | G3 (n = 127) | G4 (n = 67) | *P*-value |
|---|---|---|---|---|---|---|
| Cr (umol/l) | 904.59 ± 302.05 | 941.0 ± 291.0 | 904.0 ± 274.0 | 899.0 ± 326.0 | 894.0 ± 304.0 | 0.453 |
| Uric acid (umol/l) | 450.52 ± 118.38 | 459.0 ± 135.0 | 449.0 ± 107.0 | 444.0 ± 117.0 | 459.0 ± 127.0 | 0.753 |
| Serum potassium (mmol/l) | 4.26 ± 0.6 | 4.22 ± 0.73 | 4.25 ± 0.59 | 4.26 ± 0.60 | 4.28 ± 0.55 | 0.969 |
| Serum sodium (mmol/l) | 140.89 ± 3.18 | 141.0 ± 3.59 | 141.0 ± 3.69 | 141.0 ± 2.79 | 141.0 ± 2.82 | 0.668 |
| Carban dioxide (mmol/l) | 25.84 ± 3.41 | 26.30 ± 4.40 | 25.70 ± 3.23 | 25.60 ± 3.43 | 26.10 ± 2.99 | 0.754 |
| **Cardiac indicators** | | | | | | |
| LVEDD (mm) | 49.37 ± 6.59 | 48.30 ± 6.88 | 50.10 ± 5.53 | 43.10 ± 5.69 | 49.40 ± 6.92 | 0.001 |
| IVST (mm) | 10.13 ± 1.19 | 10.20 ± 1.31 | 10.0 ± 1.15 | 8.40 ± 1.02 | 10.10 ± 1.25 | 0.002 |
| LVPWT (mm) | 9.64 ± 1.09 | 9.81 ± 1.18 | 9.71 ± 1.12 | 8.75 ± 1.11 | 9.78 ± 1.27 | 0.018 |
| LVEF (%) | 64.85 ± 6.93 | 66.4 ± 5.35 | 64.8 ± 6.86 | 69.60 ± 6.13 | 58.50 ± 5.79 | <0.001 |

CGN, glomerulonephritis; DN, diabeticnephropathy; HN, benign hypertensive nephrosclerosis; DM, diabetes mellitus; CVD, cardiovascular disease; HP, hypertension; CRP, C-response protein; PTH, parathyroid hormone; ALP, alkaline phosphatase; ALT, alanine transaminase; AST, aspartate trans-aminase; HDL-C, high density lipoprotein-cholesterol; LDL-C, low density lipoprotein-cholesterol; BUN, blood urea nitrogen; Cr, creatinine; LVEDD, left ventricular end-diastolic diameter; IVST, interventricular septum thickness; LVPWT, left ventricularposterior wall thickness; LVEF, left ventricular ejection fraction.

### 3.4. Random survival forests

Random survival forests analysis (Fig 5) assessed the relative importance of predictors for mortality. For all-cause and CVD death, platelet count ranked first, and serum albumin ranked fifth among the most important predictors. For infection mortality, neutrophil count ranked sixth. Lymphocyte count, a component of both SII and PNI, ranked fifteenth for CVD mortality, but was still identified as important. This analysis confirmed that all four key components (platelets, neutrophils, lymphocytes, albumin) underlying the SII-PNI score play significant and complementary roles in predicting adverse outcomes in PD patients.

## 4. Discussion

This study innovatively developed and validated the SII-PNI score as a novel composite biomarker for personalized mortality risk prediction in PD patients. Our key finding was that stratifying patients based on combined SII and PNI levels effectively identifies distinct risk groups with markedly different prognosis. The high-risk group (G4: high SII + low PNI) exhibited a dramatically increased risk of all-cause, CVD and infection-related mortality, even after rigorous adjustment for numerous potential confounders. Crucially, the SII-PNI score demonstrated superior diagnostic performance compared to SII, PNI or CRP alone. By integrating readily available inflammatory and nutritional biomarkers, the SII-PNI score provides a comprehensive, clinically practical tool for personalized risk assessment in PD.

The strength of SII lies in its integration of three key immune cell types. Platelets and neutrophils drive inflammation, while lymphocytes reflect immune regulation. Elevated SII signifies an imbalance favoring pro-inflammatory states. Our findings align with previous studies linking high SII to worse outcomes in PD [17,18], but we extend this by demonstrating its potent synergy with nutritional status within the SII-PNI framework. Similarly, PNI, incorporating albumin (a marker of visceral protein stores) and lymphocytes (immune competence), is a well-established nutritional prognosticator [13,19–21]. Our results confirm that low PNI is detrimental, but more importantly, show that its combination with high inflammation (SII) defines the highest risk phenotype (G4). It is worth mentioning that, because of the low number of events (13 CVD deaths and 9 infection-related deaths), some confidence intervals are extremely wide, which may reduce the precision of these estimates. Therefore, these findings should be interpreted cautiously and validated in larger cohorts. The core innovation and clinical significance of this study lies in the combination of SII and PNI into a single, easily

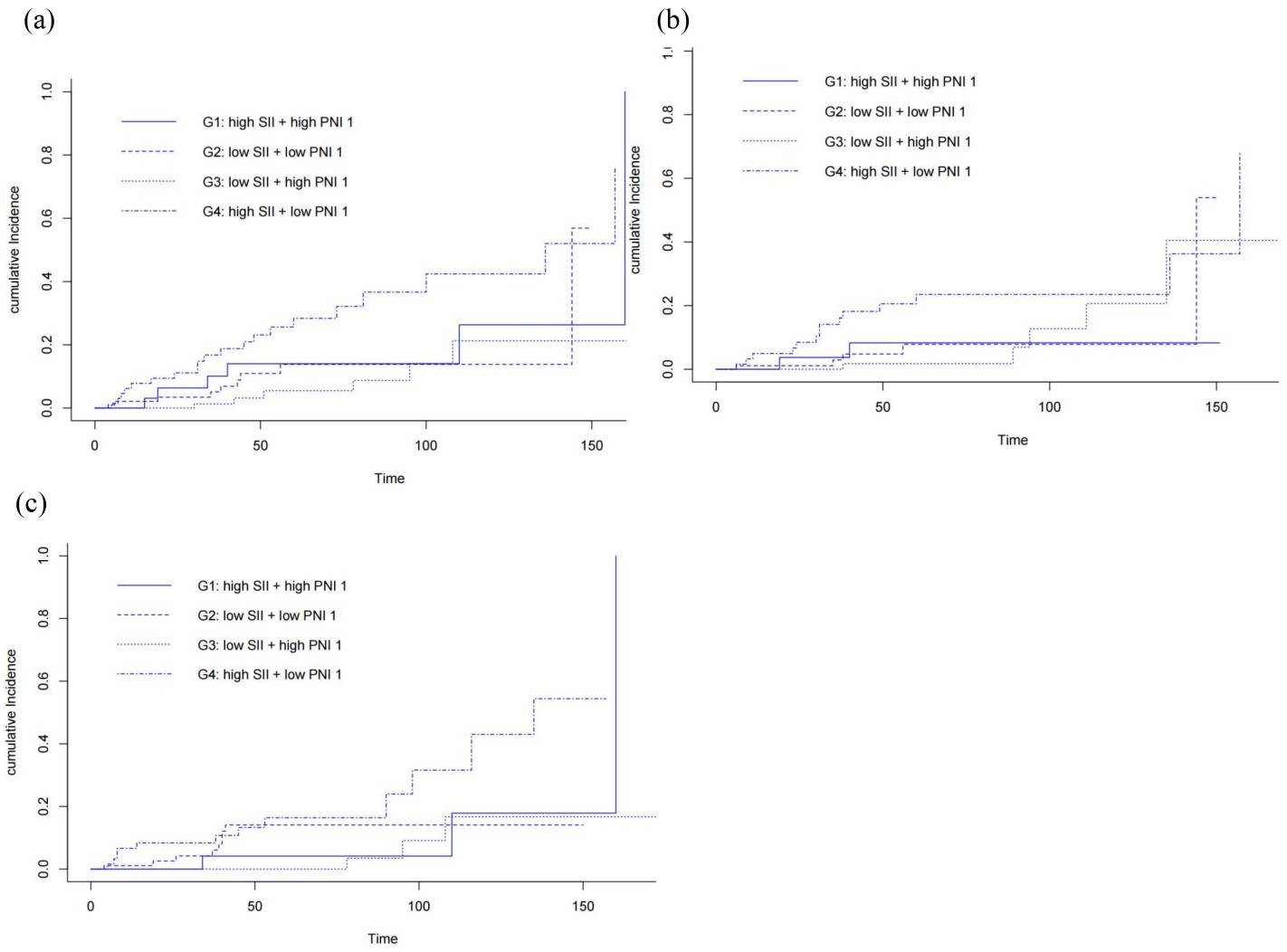

**Fig 2. Cumulative incidence rate of adverse events for (a) all-cause mortality, (b) cardiovascular mortality, (c) infection-related mortality.**

derived predictive tool. While inflammation and malnutrition are recognized intertwined culprits in PD morbidity and mortality, most prior studies focused on single or limited markers [22–24]. The SII-PNI score captures this interplay and offers a more holistic view of the inflammation-immunity-nutrition axis, focusing on synergistic effects that individual markers miss. This is evidenced by its significantly higher AUC compared to SII, PNI, or CRP alone, demonstrating that the composite score provides excellent prognostic value. Furthermore, random survival forests confirmed the significant contribution of all four cellular components (platelets, neutrophils, lymphocytes, albumin) to the predictive power, validating the biological rationale behind the composite score.

Our findings have direct implications for personalized medicine in PD. The first is risk stratification and resource allocation. The SII-PNI score provides a simple, objective method to stratify PD patients into distinct risk categories using routine laboratory data available at most centers. Patients with low risk (low SII + high PNI) have favorable inflammation and nutrition profiles, they may require standard monitoring protocols, allowing for optimized resource allocation. Patients with high risk (high SII + low PNI) face a substantially elevated mortality risk. Identification of these high-risk individuals is

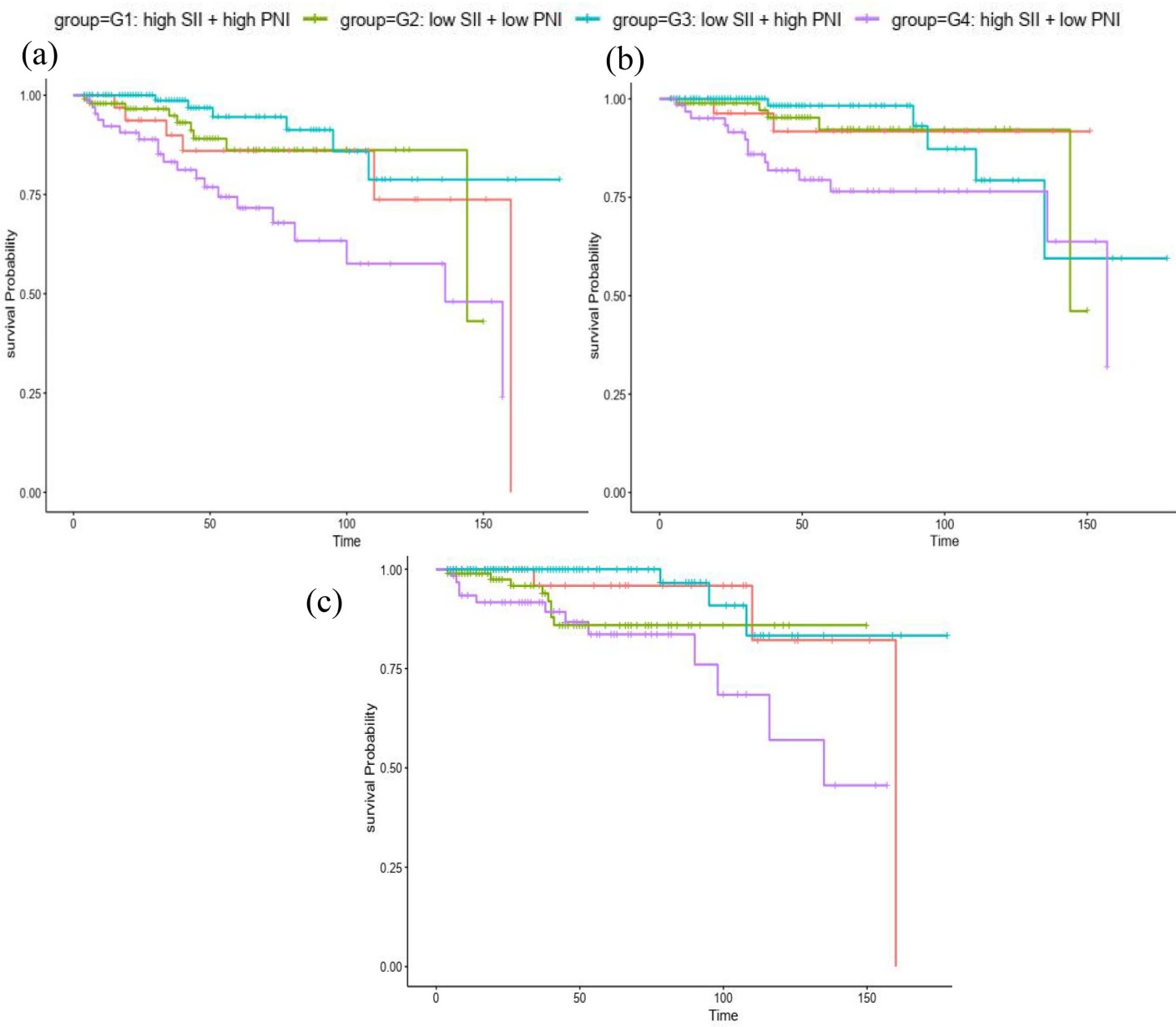

**Fig 3. Kaplan–Meier survival analysis curves of (a) all-cause, (b) cardiovascular, and (c) infection-related mortality with different groups.**

critical for triggering intensified, personalized management strategies. In terms of intermediate risk (high SII+high PNI; low SII+low PNI), these groups warrant closer observation than low-risk patients but likely less intensive intervention than the high-risk group. Further research could refine management within these subgroups. The second is clinical utility. The SII-PNI score is calculated from standard complete blood count and serum albumin, tests routinely performed in primary clinics. Its calculation is straightforward, making it highly feasible for integration into clinical workflows without additional costs or burdens. The third is dynamic monitoring. While this study used baseline averages, the SII-PNI score could

**Table 2. The relationship between different groups and all-cause, cardiovascular and infection-associated death in peritoneal dialysis patients.**

| | Unadjusted | | Model 1 | | Model 2 | | Model 3 | |
|---|---|---|---|---|---|---|---|---|
| | HR (95% CI) | *P*-value | HR (95% CI) | *P*-value | HR (95% CI) | *P*-value | HR (95% CI) | *P*-value |
| All-cause death | | | | | | | | |
| G3 | Reference | | Reference | | Reference | | Reference | |
| G1 | 0.91 (0.51-2.69) | 0.737 | 0.73 (0.39-2.21) | 0.712 | 0.81 (0.38-2.06) | 0.826 | 0.85 (0.27-2.91) | 0.704 |
| G2 | 0.68 (0.19-1.37) | 0.134 | 0.72 (0.34-1.57) | 0.056 | 0.64 (0.19-1.58) | 0.152 | 0.68 (0.19-1.27) | 0.083 |
| G4 | 3.78 (2.93-5.71) | 0.003 | 2.89 (1.95-8.46) | 0.001 | 3.35 (1.86-9.23) | 0.002 | 3.36 (1.93-8.67) | <0.001 |
| CVD death | | | | | | | | |
| G3 | Reference | | Reference | | Reference | | Reference | |
| G1 | 1.71 (0.41-6.38) | >0.900 | 1.08 (0.25-6.09) | 0.800 | 1.07 (0.14-7.13) | >0.900 | 1.07 (0.14-7.63) | >0.900 |
| G2 | 0.86 (0.13-4.73) | >0.900 | 0.61 (0.13-4.93) | 0.300 | 1.36 (0.67-4.59) | 0.300 | 1.46 (0.86-5.17) | 0.300 |
| G4 | 4.58 (2.76-15.70) | 0.003 | 3.68 (3.96-16.90) | <0.001 | 3.71 (2.78-19.90) | <0.001 | 3.74 (2.41-19.60) | <0.001 |
| Infection death | | | | | | | | |
| G3 | Reference | | Reference | | Reference | | Reference | |
| G1 | 2.29 (0.57-8.83) | 0.356 | 2.85 (0.61-10.15) | 0.251 | 2.72 (0.45-12.90) | 0.400 | 2.70 (0.49-12.80) | 0.400 |
| G2 | 0.49 (0.12-2.31) | 0.400 | 1.68 (0.12-3.89) | 0.700 | 0.69 (0.12-3.89) | 0.800 | 0.79 (0.12-3.89) | 0.800 |
| G4 | 3.41 (2.98-13.90) | 0.004 | 5.47 (2.98-22.60) | <0.001 | 4.41 (2.45-21.10) | <0.001 | 4.32 (2.58-20.40) | <0.001 |

Model 1: adjusted for age, gender, DM, HP, CVD, BMI and catheterization method.

Model 2: Model 1 + lymphocyte count.

Model 3: Model 2 + triglyceride.

potentially be used for dynamic risk monitoring. Serial measurements might track response to interventions or signal worsening status, enabling timely adjustments to care plans.

The mechanisms linking high SII and low PNI to poor outcomes are likely multifactorial. Pro-inflammatory cytokines (e.g., IL-1β, TNF-α) and growth factors released by activated neutrophils and platelets can promote endothelial dysfunction, atherosclerosis and cardiac remodeling [25]. Concurrent malnutrition and lymphopenia impair immune defenses, increasing susceptibility to infections [26]. This dysregulated inflammation-immunity-nutrition axis is central to the pathophysiology of cardiovascular-kidney-metabolic (CKM) syndrome, where organ dysfunctions interact detrimentally, creating a vicious cycle that drives adverse outcomes like accelerated CVD and infection-related death [27]. The SII-PNI score effectively captures this interconnected pathophysiology, explaining its strong prognostic power.

The strengths of our study are as follows. Firstly, we conducted a long observation period (nearly 18 years) providing robust longitudinal data. Secondly, we defined baseline as an average over 6 months post-PD start, enhancing stability and representativeness. Thirdly, we comprehensively adjusted for potential confounders in multivariate models. Finally, we focused on developing a clinically practical tool for personalized risk assessment based on routine data.The study also has some limitations. Firstly,the cut-off values for SII and PNI used in this study, although justified within our cohort, were derived from a single-center population. These thresholds are likely population-specific and may not be directly generalizable to other PD centers with different demographic or clinical characteristics. Therefore, external validation of the SII-PNI score and its cut-offs in independent, multi-center prospective cohorts is essential to confirm their general applicability and to establish universal thresholds for clinical practice. Secondly, while averaging mitigated some variability, changes in SII and PNI over time were not analyzed. Prospective studies incorporating serial SII-PNI measurements are needed to assess its dynamic predictive value and response to interventions. Thirdly, internal validation was performed, but an independent cohort is needed for full validation. Finally, while we demonstrated strong predictive value, further research is crucial to establish clear clinical decision thresholds, develop specific protocols for integrating the SII-PNI score into routine

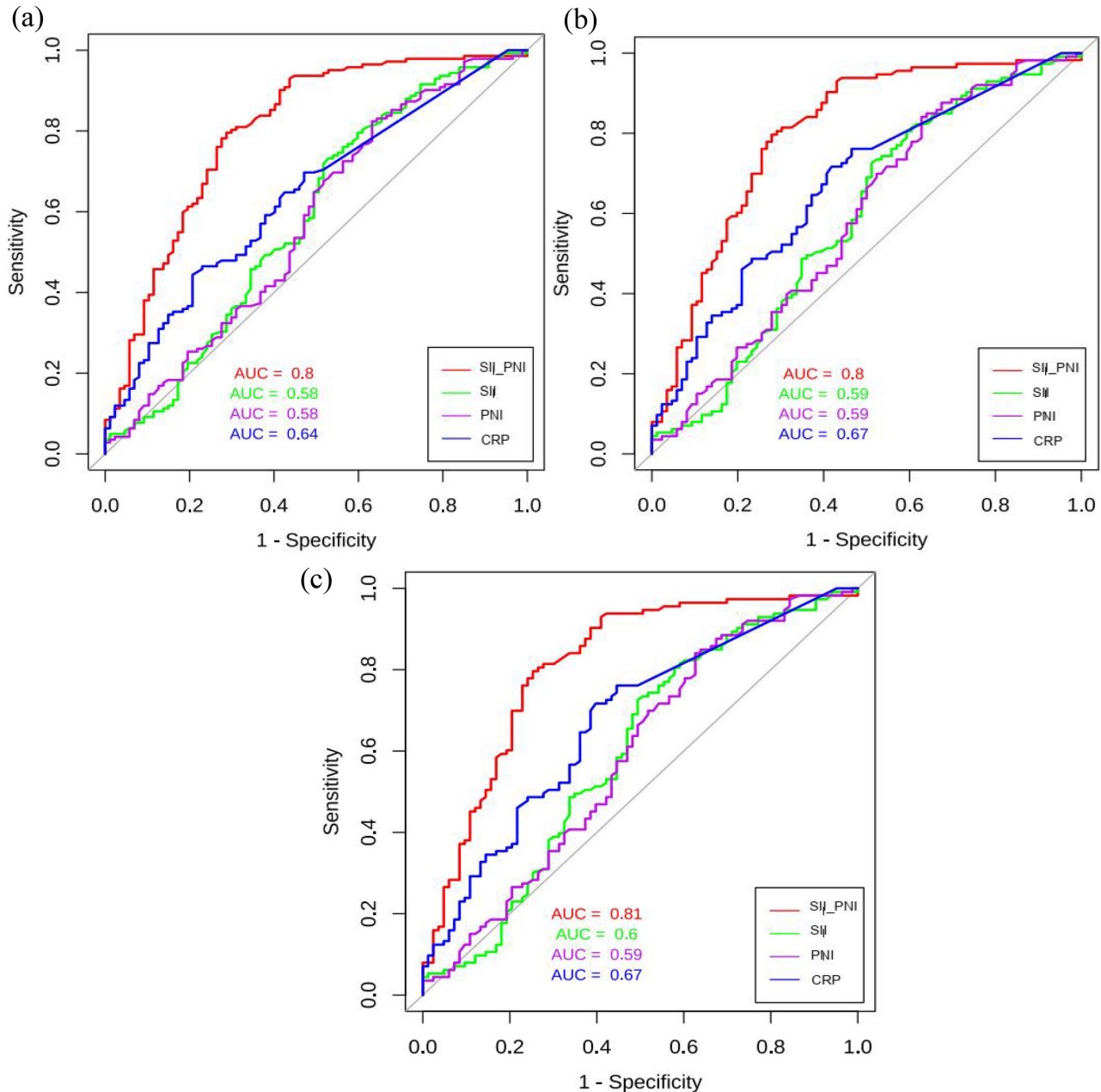

**Fig 4. The ROC curves for different markers in predicting incident (a) all-cause mortality, (b) cardiovascular mortality and (c) infection-related mortality.**

**Table 3. The diagnostic test statistics for different markers.**

|  | AUC | Sensitivity | Specificity | Positive predictive value | Negative predictive value | 95% CI | *P*-value | C-index |
|---|---|---|---|---|---|---|---|---|
| SII | 0.83 | 0.98 | 0.76 | 0.06 | 0.93 | 0.75-0.91 | <0.001 | 0.90 |
| PNI | 0.81 | 0.77 | 0.71 | 0.05 | 0.91 | 0.72-0.89 | <0.001 | 0.84 |
| SII-PNI | 0.85 | 0.70 | 0.85 | 0.06 | 0.93 | 0.77-0.92 | <0.001 | 0.92 |
| CRP | 0.73 | 0.88 | 0.67 | 0.05 | 0.99 | 0.65-0.82 | <0.001 | 0.82 |

**Fig 5. Importance of clinical predictors for predicting (a) all-cause mortality, (b) cardiovascular mortality and (c) infection-related mortality among PD patients by random survival forests.**

PD care pathways. For example, we may calculate the SII-PNI score at PD initiation and every 6 months to dynamically assess risk. A transition to a higher risk group should prompt comprehensive evaluation of inflammatory and nutritional status, followed by tailored interventions, such as dietary support, anti-inflammatory strategies. By this we can prospectively evaluate whether risk-stratified management guided by this score actually improves patient outcomes.

## 5. Conclusions

In summary, this study establishes the SII-PNI score as a novel, robust, and clinically practical composite biomarker for personalized mortality risk stratification in patients undergoing peritoneal dialysis. By integrating routinely measured indicators of SII and PNI, this score effectively identifies patients at high risk who may benefit most from intensified monitoring and targeted interventions. Its superior predictive performance, ease of calculation from standard blood tests, and biological plausibility underscore its significant potential to advance precision medicine in PD care.

## Supporting information

**S1 Data. Raw data.**
(XLSX)

## Acknowledgments

The authors thank the nursing staff and patients of the Peritoneal Dialysis Center at The Affiliated Huai'an NO.1 People's Hospital of Nanjing Medical University for their contributions to this study. We also acknowledge the support from the Huai'an Municipal Health Commission.

## Author contributions

**Conceptualization:** Min Zhou.

**Data curation:** Min Zhou, Guoyi Wang, Min Chen, Xuan Chen.

**Formal analysis:** Min Zhou, Guoyi Wang, Min Chen, Xuan Chen.

**Funding acquisition:** Min Zhou.

**Methodology:** Min Zhou.

**Project administration:** Jinwen Zhao, Yong Xu.

**Resources:** Ping Yue, Hua Lin.

**Supervision:** Ping Yue, Hua Lin, Yong Xu.

**Writing – original draft:** Min Zhou, Guoyi Wang.

**Writing – review & editing:** Jinwen Zhao, Yong Xu.

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
