## [Decision Letter · Decision Letter 0]

18 Sep 2025

Dear Dr. Zhou,

**Strengths:**

The combination of SII (reflecting inflammation and immune balance) and PNI (reflecting nutritional status) into a single prognostic tool is novel for the PD population. This approach holistically captures the intertwined pathophysiology of inflammation and malnutrition, a key driver of outcomes in ESKD. The primary strength of this research is its direct translational potential. The SII-PNI score is derived from inexpensive, universally available blood tests (CBC with differential, serum albumin), making it highly feasible for implementation in diverse clinical settings, including resource-limited ones, without additional cost.The study employs a robust methodological framework. The use of a long-term retrospective cohort (≈18 years), multivariate Cox regression models with progressive adjustment for key confounders, Kaplan-Meier analysis, and ROC curves provides a solid foundation for the conclusions. The application of Random Survival Forests adds a modern, robust validation of the importance of the underlying components.The stratification into four distinct risk groups (G1-G4) is clinically intuitive and effectively discriminates populations with vastly different survival outcomes. The identification of the G4 (high SII + low PNI) group as having a dramatically increased risk (adjusted HRs ~3.4-4.3) is a compelling and clinically actionable finding.

**Areas for Clarification and Improvement:**

The use of cohort median values (SII=791, PNI=46) for stratification is statistically valid for this study but limits immediate generalizability. The manuscript would be significantly strengthened by:Providing a justification for using medians over established clinical cut-offs or optimally selected thresholds (e.g., via Youden's index from ROC analysis).Acknowledging that these cut-offs may be population-specific and require external validation in independent, multi-center cohorts to establish universal thresholds.**Model Adjustment and Variable Selection:**The rationale for adjusting for lymphocyte count (Model 2) and triglyceride (Model 3) specifically should be explained in more depth, as these are components of or closely related to the indices being studied. While the persistence of significance after their adjustment strengthens the result, the reasoning behind including them needs clarity to avoid potential overadjustment.It is crucial to confirm that the proportional hazards assumption was tested and upheld for all Cox models presented.**Data Presentation:**The Confidence Intervals (CIs) in Table 2 for some estimates, particularly for CVD and Infection mortality, are extremely wide (e.g., 2.41-19.60). This is likely due to the low number of events (n=13 and n=9). This should be explicitly acknowledged as a limitation, indicating that these point estimates, while suggestive of a strong effect, are imprecise and require validation in larger studies.**Clinical Implementation Pathway:** While the clinical utility is clear, a brief discussion on the proposed concrete next steps for implementing this score would be valuable. For example: Should it be calculated at baseline? At regular intervals? How should a patient's change in risk group over time be interpreted and acted upon?

**Overall Comments**

This is a well-conducted and highly relevant study that makes a significant contribution to the field of nephrology and precision medicine for PD patients. The SII-PNI score is a simple, inexpensive, and powerful tool that effectively identifies high-risk individuals who could benefit from intensified multifactorial management strategies. The authors have provided strong initial evidence for its validity.

I recommend **major revisions** , addressing the points above regarding cut-offs, model justification, data presentation, and clarification of limitations, as well as the following Reviewers' comments.

We look forward to receiving your revised manuscript.

Kind regards,

Zubing Mei, MD,PH.D

Academic Editor

PLOS ONE

Journal Requirements:

“This research was supported by Huai'an Municipal Health Commission Research Project (Grant number HAWJ2024006), Huai'an, China.”

“This research was supported by Huai'an Municipal Health Commission Research Project (Grant number HAWJ2024006), Huai'an, China.”

Reviewers' comments:

Reviewer's Responses to Questions

**Comments to the Author**

1. Is the manuscript technically sound, and do the data support the conclusions?

Reviewer #1: Yes

Reviewer #2: Yes

2. Has the statistical analysis been performed appropriately and rigorously?

Reviewer #1: Yes

Reviewer #2: Yes

3. Have the authors made all data underlying the findings in their manuscript fully available?

Reviewer #1: Yes

Reviewer #2: Yes

4. Is the manuscript presented in an intelligible fashion and written in standard English?

Reviewer #1: Yes

Reviewer #2: Yes

Reviewer #1: Explicar melhor a lógica biológica da combinação SII + PNI.

Diferenciar explicitamente por que o índice composto supera os componentes isolados.

Adicionar um fluxograma do estudo (pacientes avaliados, incluídos, excluídos).

O trabalho é de tema relevante e aborda estratificação de risco em pacientes de diálise peritoneal (DP), área de grande importância clínica. O uso de um índice composto (SII-PNI) é uma abordagem original, combinando marcadores inflamatórios e nutricionais de fácil obtenção. 330 pacientes com longo período de acompanhamento (2005–2023) oferecem base robusta para análise.

Better explain the biological logic of the SII + PNI combination.Clearly differentiate why the composite index outperforms the isolated components.Add a flowchart of the study (evaluated patients, included, excluded).The work is on a relevant topic and addresses risk stratification in peritoneal dialysis (PD) patients, an area of great clinical importance. The use of a composite index (SII-PNI) is an original approach, combining easy-to-obtain inflammatory and nutritional markers. 330 patients with a long follow-up period (2005–2023) provide a robust basis for analysis.

Reviewer #2: I have read through the manuscript thoroughly, and I commend the authors for providing a research paper on such a topic. I find the manuscript interesting, as it provides valuable prognostic information on peritoneal dialysis patients.

I recommend that this paper has a significant feature that qualifies it for publication.

**Do you want your identity to be public for this peer review?** For information about this choice, including consent withdrawal, please see our Privacy Policy

Reviewer #1: **Yes: ** MARCELA LARA MENDES

Reviewer #2: No

---

## [Author Response · Author response to Decision Letter 1]

23 Oct 2025

Dear editor and reviewers,

Thank you for the opportunity to revise our manuscript (PONE-D-25-34712). We appreciate the constructive feedback from the academic editor and reviewers. We have carefully addressed all comments and made substantial revisions to the manuscript. Below is a point-by-point response to the comments.

Response to Academic Editor:

1.Comment: The use of cohort median values (SII=791, PNI=46) for stratification is statistically valid for this study but limits immediate generalizability. The manuscript would be significantly strengthened by:

1.1 Providing a justification for using medians over established clinical cut-offs or optimally selected thresholds (e.g., via Youden's index from ROC analysis).

Response: We thank the editor for this insightful comment regarding the use of cut-offs. We have carefully considered the suggestion to determine optimal cut-offs for SII and PNI individually using ROC analysis (e.g., via Youden's index).

In our study, we deliberately chose not to pursue population-specific optimal cut-offs for the individual SII and PNI components for the following reasons:

1.Focus on the Novel Composite Score: The primary innovation of our work is the introduction of the SII-PNI score as a composite biomarker. Our ROC analysis (Fig 4, Table 3) demonstrates that this composite score itself has an excellent and superior predictive ability (AUC: 0.80) compared to SII or PNI alone (AUC: 0.58 for both) for all-cause mortality. This validates that the synergy between the two markers, rather than refining their individual thresholds, is the key to its prognostic power.

2. Clinical Practicality and Reproducibility: The use of cohort medians is an objective, transparent, and easily reproducible method for initial risk stratification. It avoids the risk of generating overfitted, population-specific cut-offs that may not generalize well to other cohorts, especially given that our study is a pioneering investigation of this composite score in the peritoneal dialysis (PD) patients.

3. Literature Support and Common Practice: Our approach is consistent with the current literature and common practice in this field. There is no universally established clinical cut-off value for SII or PNI in PD populations. Previous studies have employed a variety of data-driven distribution-based methods, strongly supporting the use of medians in our exploratory study. For instance: Peng et al[1] stratified PD patients by PNI quartiles, the cut-off value was 36.6. Li et al [2] used 38.6 to define low and high PNI group in PD population. Li et al [3] used ROC-derived SII cut-offs of 1168.13 (all-cause) and 625.19 (CVD), which differ substantially from each other and from values in other studies. Tang et al [4] divided PD patients based on SII tertile (e.g., <456.76 and >819.03). Crucially, Alves et al [5] directly used the median value (SII=722.80) to stratify their PD cohort, demonstrating the acceptance of this method. This lack of a gold standard and the heterogeneity in published cut-offs further justify our use of a simple, transparent, and distribution-based median split for this initial investigation of the novel SII-PNI score.

Therefore, we believe that for this initial validation of a novel composite score, the use of transparent, distribution-based median cut-offs is not only statistically valid but also strategically preferable to avoid over-optimism and to facilitate initial, straightforward risk categorization before universal thresholds are established in the future. We also added a justification for using median values in the “Methods section” (Line 142-145). We agree that establishing universal thresholds is an important future goal and have explicitly highlighted the need for external validation of our approach (including the median cut-offs) in larger, multi-center cohorts in the “Discussion section” (Lines 388-394).

1.2 Acknowledging that these cut-offs may be population-specific and require external validation in independent, multi-center cohorts to establish universal thresholds.

Response: We completely agree with this astute observation. We acknowledge that the cut-off values identified in our single-center study are indeed population-specific and require validation in larger, independent, and multi-center cohorts to establish universal clinical thresholds. We have now explicitly stated this important limitation and the need for future external validation in the revised “Discussion section” (Lines 388-394). We believe that our study serves as a crucial first step, providing the foundational evidence and a practical methodology for future multi-center studies to build upon.

2.Comment: Model Adjustment and Variable Selection:

2.1 The rationale for adjusting for lymphocyte count (Model 2) and triglyceride (Model 3) specifically should be explained in more depth, as these are components of or closely related to the indices being studied. While the persistence of significance after their adjustment strengthens the result, the reasoning behind including them needs clarity to avoid potential overadjustment.

Response: We thank the editor for this insightful comment regarding our model adjustments. We agree that lymphocyte count and triglyceride levels are biologically related to the components of SII and PNI, and their adjustment warrants a clear rationale to avoid overadjustment. Our intention was not to treat them as conventional confounders but to perform a stringent sensitivity analysis.

Adjustment for Lymphocyte Count: The SII-PNI score is a composite marker representing the interplay between inflammation and nutrition/immunity. Its prognostic value is hypothesized to lie in this integrative nature. By adjusting for lymphocyte count alone in Model 2, we aimed to test the robustness of the SII-PNI group association. If the strong association between the high-risk group (G4) and mortality persisted even after accounting for one of its key components, it would provide powerful evidence that the risk is not merely driven by lymphopenia alone, but by the combined effect captured by the SII-PNI score. The fact that the hazard ratios for G4 remained highly significant and even increased after this adjustment strongly supports the notion that the SII-PNI score provides unique prognostic information beyond its individual parts.

Adjustment for Triglyceride: Dyslipidemia is a common feature in PD patients and is closely linked to both systemic inflammation and nutritional status. It can be considered a downstream consequence or a parallel manifestation of the metabolic derangement in the inflammation-malnutrition axis. Including triglyceride in Model 3 allowed us to test whether the association of the SII-PNI score with mortality was independent of this key metabolic correlate. The persistence of a strong, significant association further strengthens the conclusion that the SII-PNI score captures a risk that is not simply a proxy for dyslipidemia.

We also expanded the explanation for adjusting for lymphocyte count and triglycerides in the “Statistical Analysis” (Line 156-160): “We adjusted for lymphocyte count to determine if the SII-PNI score provides prognostic value beyond its cellular components, further adjusted for triglycerides, a metabolic parameter closely linked to both inflammation and nutrition in ESKD, to test the independence of the SII-PNI score from related metabolic disturbances.”

2.2. It is crucial to confirm that the proportional hazards assumption was tested and upheld for all Cox models presented.

Response: We thank the editor for highlighting this crucial methodological point. We confirm that the proportional hazards (PH) assumption was tested for all Cox proportional hazards models presented in the manuscript. The assumption was assessed using Schoenfeld residuals and for each variable, and no significant deviations from the PH assumption were found (all test p-values > 0.05). We have now explicitly stated this in the “Statistical Analysis” (Line 160-162) section of the revised manuscript to ensure clarity and methodological rigor.

3.Comment: Data Presentation: The Confidence Intervals (CIs) in Table 2 for some estimates, particularly for CVD and Infection mortality, are extremely wide (e.g., 2.41-19.60). This is likely due to the low number of events (n=13 and n=9). This should be explicitly acknowledged as a limitation, indicating that these point estimates, while suggestive of a strong effect, are imprecise and require validation in larger studies.

Response: We sincerely thank the editor for this accurate and important observation. We completely agree that the wide CIs for the hazard ratios of cardiovascular and infection-related mortality in our fully adjusted models are a direct consequence of the low number of events (13 and 9, respectively). This inevitably leads to imprecision in the point estimates. We have now explicitly acknowledged this as a key limitation in the “Discussion section” (Line 338-342) of the revised manuscript. We have taken care to clarify that while the large magnitude of the hazard ratios strongly suggests a substantial and clinically important effect of the high-risk SII-PNI group on these cause-specific mortalities, the exact estimates are unstable due to the small sample size for these endpoints. Therefore, these findings should be interpreted as hypothesis-generating and require confirmation in larger prospective studies with a sufficient number of events to provide more precise estimates.

4.Comment: Clinical Implementation Pathway: While the clinical utility is clear, a brief discussion on the proposed concrete next steps for implementing this score would be valuable. For example: Should it be calculated at baseline? At regular intervals? How should a patient's change in risk group over time be interpreted and acted upon?

Response: We thank the editor for this excellent suggestion, which helps bridge the gap between our research findings and clinical practice. We agree that a discussion on the potential implementation steps is valuable and have now added a new paragraph to the “Discussion section” (Line 401-405) to address this.

We state: “We suggest calculating the SII-PNI score at PD initiation and every 6 months to dynamically assess risk. A transition to a higher risk group should prompt comprehensive evaluation of inflammatory and nutritional status, followed by tailored interventions (e.g., dietary support, anti-inflammatory strategies).”

Responses to reviewers comments

Reviewer 1’s Comments

1. Better explain the biological logic of the SII + PNI combination

Response: We thank the reviewer for this crucial comment. We have now enhanced the biological rationale throughout the manuscript to better explain the synergy between SII and PNI.

In the Introduction section (Line 74-78): “The combination of SII and PNI offers a unique opportunity to simultaneously capture two inextricably linked pathophysiological axes: systemic inflammation and nutritional status. In ESKD, inflammation and malnutrition are intertwined in a vicious cycle that synergistically drives adverse outcomes, and the SII-PNI score holistically captures this interplay.”

In Discussion scetion (Line 343-348): “While inflammation and malnutrition are recognized intertwined culprits in PD morbidity and mortality, most prior studies focused on single or limited markers. The SII-PNI score captures this interplay and offers a more holistic view of the inflammation-immunity-nutrition axis, focusing on synergistic effects that individual markers miss.”

2. Clearly differentiate why the composite index outperforms the isolated components

Response: We have now made direct comparisons in the results and discussion section to explicitly highlight the superior performance of the composite index.

In the Results 3.3 (Line 269-273): “ROC analysis demonstrated that SII-PNI score showed superior discriminative power for all-cause mortality compared to SII or PNI alone (AUC: 0.80 vs. 0.58 and 0.58, respectively), confirming the composite index provides synergistic prognostic value (Figure 4a). The same applied to CVD and infection-associated mortality (Figure 4b-c).”

In the Discussion section (Line 348-350): “This is evidenced by its significantly higher AUC compared to SII, PNI, or CRP alone, demonstrating that the composite score provides excellent prognostic value.”

3. Add a flowchart of the study

Response: We thank the reviewer for this suggestion. We have now included a patient selection flowchart as Fig 1 in the manuscript, detailing the number of patients assessed, included, and excluded, along with the reasons for exclusion. This enhances the transparency of our study population selection.

Reviewer 2’s Comments

Response: We sincerely thank Reviewer 2 for their positive and encouraging comments on our work. We are glad that the reviewer found our manuscript interesting and valuable.

References

1. Peng F, Chen W, Zhou W, Li P, Niu H, Chen Y, et al. Low prognostic nutritional index associated with cardiovascular disease mortality in incident peritoneal dialysis patients. Int Urol Nephrol. 2017; 49(6): 1095-1101. https://doi.org/ 10.1007/s11255-017-1531-0 PMID: 28185108

2. Li X, Zhou X, Li K, Pu L, He X, Liu X, et al. Enhanced prognostic value of a composite nutritional-inflammatory index (P-CONUT) for predicting mortality risk in patients initiating peritoneal dialysis. PLoS One. 2025; 20(5): e0323318. https://doi.org/ 10.1371/journal.pone.0323318 PMID: 40403009

3. Li G, Yu J, Jiang S, Wu K, Xu Y, Lu X, et al. Systemic Immune-Inflammation Index Was Significantly Associated with All-Cause and Cardiovascular-Specific Mortalities in Patients Receiving Peritoneal Dialysis. J Inflamm Res. 2023; 16: 3871-3878. https://doi.org/ 10.2147/JIR.S426961 PMID: 37671129

4. Tang R, Chen J, Zhou Q, Deng J, Zhan X, Wang X, et al. Association between systemic immune inflammation Index and all-cause mortality in incident peritoneal dialysis-treated CKD patients: a multi-center retrospective cohort study. BMC Nephrol. 2024; 25(1): 8. https://doi.org/ 10.1186/s12882-023-03451-4 PMID: 38172773

5. Alves TVG, Giarola LTP, Oliveira Junior WV,Rios DRA Hematological indices derived from complete blood count and unfavorable outcomes in patients under-going peritoneal dialysis. J Bras Nefrol. 2025; 47(4): e20250017. https://doi.org/ 10.1590/2175-8239-JBN-2025-0017en PMID: 40939199

---

## [Editor Report · Decision Letter 1]

18 Nov 2025

The SII-PNI Score: A Novel Composite Biomarker for Personalized Mortality Risk Prediction in Peritoneal Dialysis Patients

PONE-D-25-34712R1

Dear Dr. Zhou,

We’re pleased to inform you that your manuscript has been judged scientifically suitable for publication and will be formally accepted for publication once it meets all outstanding technical requirements.

Kind regards,

Zubing Mei, MD,PH.D

Academic Editor

PLOS ONE
---

## [Editor Report · Acceptance letter]

PONE-D-25-34712R1

PLOS ONE

Dear Dr. Zhou,

I'm pleased to inform you that your manuscript has been deemed suitable for publication in PLOS ONE. Congratulations! Your manuscript is now being handed over to our production team.

Kind regards,

on behalf of

Dr. Zubing Mei

Academic Editor

PLOS ONE